# Increased Oxidative Stress Markers in Acute Ischemic Stroke Patients Treated with Thrombolytics

**DOI:** 10.3390/ijms232415625

**Published:** 2022-12-09

**Authors:** Hanna Pawluk, Renata Kołodziejska, Grzegorz Grześk, Alina Woźniak, Mariusz Kozakiewicz, Agnieszka Kosinska, Mateusz Pawluk, Elżbieta Grzechowiak, Jakub Wojtasik, Grzegorz Kozera

**Affiliations:** 1Department of Medical Biology and Biochemistry, Faculty of Medicine, Collegium Medicum in Bydgoszcz, Nicolaus Copernicus University in Toruń, Karłowicza 24, 85-092 Bydgoszcz, Poland; 2Department of Cardiology and Clinical Pharmacology, Faculty of Health Sciences, Collegium Medicum in Bydgoszcz, Nicolaus Copernicus University in Toruń, Ujejskiego 75, 85-168 Bydgoszcz, Poland; 3Division of Biochemistry and Biogerontology, Department of Geriatrics, Faculty of Health Sciences, Collegium Medicum in Bydgoszcz, Nicolaus Copernicus University in Toruń, Dębowa 3, 85-626 Bydgoszcz, Poland; 4Centre for Languages & International Education, University College London, 26 Bedford Way, London WC1H 0AP, UK; 5Department of Neurology, Faculty of Medicine, Collegium Medicum in Bydgoszcz, Nicolaus Copernicus University in Toruń, Marii Skłodowskiej Curie 9, 85-094 Bydgoszcz, Poland; 6Statistical Analysis Centre, Nicolaus Copernicus University in Toruń, Chopin 12/18, 87-100 Toruń, Poland; 7Centre of Medical Simulations, Medical University of Gdańsk, Dębowa 25, 80-204 Gdańsk, Poland

**Keywords:** ischemic stroke, cerebral thrombolysis, melatonin, carbonyl groups, neurological assessment

## Abstract

One of the most common neurological disorders involving oxidative stress is stroke. During a stroke, the balance of redox potential in the cell is disturbed, and, consequently, protein oxidation or other intracellular damage occurs, ultimately leading to apoptosis. The pineal gland hormone, melatonin, is one of the non-enzymatic antioxidants. It not only modulates the perianal rhythm but also has anti-inflammatory properties and protects against stress-induced changes. The focus of this research was to evaluate the concentration of the carbonyl groups and melatonin metabolite in time in patients with acute ischemic stroke that were treated with intravenous thrombolysis. This included a comparison of the functional status of patients assessed according to neurological scales with the control sample comprising healthy people. The studies showed that the serum concentrations of carbonyl groups, which were elevated in patients with ischemic stroke (AIS) in comparison to the control samples, had an impact on the patients’ outcome. A urine concentration of the melatonin metabolite, which was lower in patients than controls, was related to functional status after 24 h from cerebral thrombolysis. It shows that determination of carbonyl groups at different time intervals may be an important potential marker of protein damage in patients with AIS treated with cerebral thrombolysis, and that impaired melatonin metabolism induces a low antioxidant protection. Thus, due to the neuroprotective effects of melatonin, attention should also be paid to the design and conduct of clinical trials and hormone supplementation in AIS patients to understand the interactions between exogenous melatonin and its endogenous rhythm, as well as how these relationships may affect patient outcomes.

## 1. Introduction

The development of several diseases and permanent damages to the body are caused by the disruption of the oxidative-antioxidant homeostasis of the organism. One of the most common free radical neurological disorders is stroke.

Oxidative stress is one of the causes of the pathogenesis of ischemic nerve cell damage mediated by calcium levels [1,2]. Disorders of calcium homeostasis and its uncontrolled increase in nerve cells activate the production of reactive oxygen species (ROS). ROS can also be constitutively produced by damaged mitochondria and by the activation of cell receptors that transduce signals from inflammatory mediators.

In patients with AIS, ROS are produced during the ischemic and reperfusion phases, leading to brain damage. ROS are responsible for triggering the oxidation of protein that causes the peroxidation of polyunsaturated fatty acids when the equilibrium of the redox potential in the cell is disturbed [3,4,5,6]. Reactive oxygen species can also lead to other intracellular damage, i.e., inactivation of enzymes, structural changes in carbohydrate molecules and, consequently, to apoptosis [5,7,8] (Figure 1).

The oxidation of amino acid residues leads to the formation of relatively stable carbonyl groups that can act as qualitative and quantitative markers enabling the evaluation of protein oxidative damage [9,10]. Protein carbonylation is recognised as the non-enzymatic addition of aldehydes or ketones to specific amino acid residues. However, the literature lacks research related to the role of protein carbonyls in AIS [11,12]. Moreover, there are only a few reports on the dynamic changes of this oxidative stress marker and the contribution of antioxidants to AIS. One of the important non-enzymatic antioxidants is melatonin, the pineal gland hormone, which in addition to modulating the circadian rhythm, also has anti-inflammatory significance [5,8,13]. Melatonin protects against stress-induced changes by scavenging free radicals, such as the hydroxyl radical (•OH), peroxynitrate(III) anion (ONOO^−^), and peroxynitric(III) acid (ONOOH) or its activated form ONOOH* and against singlet oxygen toxicity [6,14]. Melatonin also has neuroprotective and anti-angiogenic properties [15]. It exhibits several properties that counteract negative biochemical reactions that occur during acute ischemia of the brain tissue [5,16].

This study focused on the evaluation of patients with AIS treated with intravenous thrombolysis. The process of selecting patients for this research was limited by the age, neurological deficit, and time of biological material collection. Moreover, the determinations of oxidative stress biomarkers presented in those studies are performed in a wide-time window or from blood serum collected only after the completion of thrombolytic therapy. The influence of initial markers of oxidative stress on the prognosis of intravenous thrombolytic therapy is unknown, although the authors of the publication raise the issue of the application of antioxidants in the treatment and prevention of stroke [17].

This study aimed to evaluate the progressive profile of the melatonin metabolite, 6-hydroxymelatonin sulphate (6-SM), in the urine, as well as to assess carbonyl groups in the blood of patients with acute ischemic stroke treated with intravenous thrombolysis. The research also focused on the comparison of the obtained results with the functional status of patients assessed according to the neurological scales with the control sample.

## 2. Results

### 2.1. Clinical Picture of Stroke Patients Treated with Intravenous Thrombolysis

The studied sample, which consisted of 123 patients, was divided into four groups based on the time at which the date was collected (1st Group—admission to hospital; 2nd Group—discharge from hospital; 3rd Group—after three months; 4th Group—after a year since the stroke). The date was collected from 122 patients in Group 1 and 119 in Group 2, and from 111 and 95 patients after three months and after a year after the stroke, respectively. Each group was assessed based on the modified Rankin Scale (mRS). In 26 patients from Group 1 (21%), 90 from Group 2 (76%), 84 from Group 3 (76%), and 78 (82%) from Group 4 favourable mRS was observed. A total of 15 patients died, including eight within 3 months of the stroke because of cerebral haemorrhage or cerebral oedema, as well as cardiopulmonary failure, sepsis, and pneumonia. A previous history of hypertension was detected in 103 (83.7) patients, coronary artery disease in 27 (22.1%); diabetes mellitus in 50 (41.0); atrial fibrillation in 15 (12.3%), hyperlipidaemia in 13 (10.6%), in 6 (4.9%) and 3 (2.5%) were diagnosed with gout and renal failure.

In the study group, 36 (29.0%) patients were administered statins and three (2.5%) were given anticoagulants. Patients in the age range between 35 and86 (median 59; *p* = 0.002) who had a favourable Rankin score demonstrated smaller mRS scores both at discharge (*p* < 0.001), after 3 months (*p* < 0.001), and after a year (*p* = 0.003) since the stroke. They also had small neurological changes according to the National Institutes of Health Stroke Scale (NIHSS) at admission and discharge (*p* < 0.001 and *p* < 0.001). They had less frequently experienced infections (*p* = 0.007), took antibiotics (*p* = 0.007), and had coronary heart disease (*p* = 0.014), they also had lower mean systolic blood pressure values (*p* = 0.002) as summarized in Table 1.

### 2.2. Melatonin and Carbonyl Group Levels before and after Intravenous Thrombolysis

Biomarker values were grouped according to the patients’ functional status (favourable and unfavourable mRS scores) at admission, discharge, and three months and one year after stroke.

The median concentration of serum carbonyl groups determined during the first 4.5 h (prior rt-PA use) was lower in the subgroup of patients with a favourable functional result (mRS: 0–2 pts) compared to the group of patients with an unfavourable functional result (mRS: 3–6 pts) assessed at admission, discharge, after 3 months, and after a 1 year. Levels of this biomarker <4.5 h were 180.58 with IQR 118.95–219.67 vs. 255.15 with IQR 183.21–294.12 (*p* = 0.002), 202.85 with IQR 157.72–267.31 vs. 274.65 with IQR 234.80–301.28 (*p* = 0.007), 203.23 with IQR 164.19–274.65 vs. 259.63 with IQR 230.33–306.63 (*p* = 0.012), 197.85 with IQR 157.72–258.25 vs. 291.47 with IQR 244.18–361.20, U/mL (*p* = 0.005), respectively.

There were also statistically significant differences in the level of melatonin metabolite marked after 24 h in the case of patients with favourable (mRS: 0–2 pts) and unfavourable (mRS: 3–6 pts) functional results at the time of admission (31.00 with IQR 8.66–36.60 vs. 8.02 with IQR 4.68–19.10, *p* = 0.010) that are summarised in Table 2.

### 2.3. The Levels of Biomarkers Compared to the Control Group

The levels of the melatonin metabolite and carbonyl groups were also evaluated and compared with the concentrations observed in the control group. Melatonin metabolite concentration measured <4.5 h on the 1st and 7th day after stroke was statistically lower than in the group of healthy people (medians 6.41, 9.34 and 9.61 vs. 37.85 pg/mL; *p* < 0.001, respectively), and the concentration of carbonyl groups was higher (222.02 vs. 144.18; *p* < 0.001, 355.56 vs. 144.18; *p* <0.001, 195.08 vs. 144.18 U/mL; *p* = 0.032, respectively), were observed as shown in Table 3.

### 2.4. The Statistical Connection between the Level of Biomarkers and the Severity of Neurological Symptoms

Statistically significant weak positive correlations were found between the measurement of carbonyl groups marked in time <4.5 h with NIHSS scales at admission and discharge (R = 0.26, *p* = 0.04; R = 0.28, *p* = 0.03, respectively) and with mRS scales on admission, on discharge, after 3 months, and after one year (R = 0.34, *p* < 0.01; R = 0.37, *p* < 0.01; R = 0.34, *p* = 0.04; R = 0.30, *p* < 0.01, respectively). Additionally, a weak negative correlation was demonstrated between the concentration of melatonin metabolite determined on the first day and the mRS scale on admission (R = −0.32, *p* = 0.01)—Table 4. There was also a correlation between the age of AIS patients and the carbonyl marker (R = 0.27, *p* = 0.01).

### 2.5. Correlations between the Levels of Biomarkers

The studies showed connections between MEL and CG concentrations measured at different time intervals. A significant statistical correlation between melatonin metabolite concentration and carbonyl groups in patients with AIS was detected as illustrated by Figure 1. Such a relationship was found between 6-hydroxymelatonin sulphate measured in <4.5 h and on the first day after the stroke, and the concentration of carbonyl groups was determined after one week (R = 0.42, R = 48; *p* < 0.05). There was also a weak negative correlation between the levels of melatonin metabolite and carbonyl groups during <4.5 h from the stroke (R = −0.26).

However, there was no effect of taking statins on the concentration values of the tested biomarkers.

### 2.6. ROC Curves

Serum carbonyl concentrations <4.5 h showed good sensitivity and specificity in predicting patient outcome on admission (AUS = 0.721), after discharge (AUC = 0.703), and after 3 months (AUC = 0.694), as well as after a 1 year (AUC = 0.797). The cut-off points for these curves are 236.38 U/mL (sensitivity 82.6%, specificity 57.9%), 256.05 U/mL (sensitivity 69.0%, specificity 70.0%), 198.22 U/mL (sensitivity 47.2%, specificity 89.5%), 198.22 U/mL (sensitivity 50.9%, specificity 100%), respectively. The results are presented in Figure 2.

### 2.7. The Level of Biomarkers Depending on Subtypes of Strokes

The concentration of carbonyl groups during <4.5 h and on the first day after the stroke showed statistically significant differences between the values of this biomarker in patients with lacunar cerebral infarcts (LACI) and patients with posterior circulation infarcts (POCI) and partial anterior circulation infarcts (PACI) (176.89 with IQR 134.82–211.50 vs. 292.84 with IQR 263.30–344.61, *p* < 0.001; 176.89 with IQR 134.82–211.50 vs. 282.67 with IQR 244.18–323.49, *p* < 0.001; 268.42 with IQR 213.02–331.31 vs. 464.87 with IQR 397.84–632.60, *p* = 0.006; 268.42 with IQR 213.02–331.31 vs. 432.49 with IQR 311.32–660.82 U/mL, *p* = 0.013). The concentration of carbonyl groups in patients with lacunar strokes was statistically significantly lower compared to other types of stroke.

In patients with LACI stroke, MEL concentrations measured in time <4.5 h were statistically higher than in patients with POCI and PACI stroke subtypes (7.27 with IQR 5.31–22.29 vs. 5.32 with IQR 3.55–6.84 pg/mL, *p* = 0.008; 7.27 with IQR 5.31–22.29 vs. 3.73 with IQR 2.60–5.61 pg/mL, *p* < 0.001, respectively). Additionally, differences in MEL concentrations for LACI and PACI stroke were observed, which were assessed within 24 h (11.84 with IQR 7.05–34.29 vs. 4.72 with IQR 3.65–8.02 pg/mL, *p* = 0.02) after the stroke. A summary of the obtained results is presented in Table 5 and Figure 3.

### 2.8. The Dependence of Mortality on Biomarker Concentrations

The biomarker concentrations were linked to the mortality of AIS patients. The increased mortality in patients after ischemic stroke was found if the CG concentration assessed <4.5 h exceeded the value of 250 U/mL and after 24 h above the value of 350 U/mL. In the case of measuring carbonyl after 7 days, the increase in mortality is not as rapid as in the case of previous measurements, which may be related to the sample size considered in the study.

An increase in mortality was also observed for the MEL levels measured <4.5 h, during 24 h and on the 7th day (<10 pg/mL determined <4.5 h and within 24 h; <15 pg/mL determined on the 7th day) as presented in Figure 4.

## 3. Discussion

Ischemic stroke occurs when the oxygen uptake by brain cells is reduced, and oxidative damage is recognized at various stages of this neuro-mechanism [1,2]. Under these circumstances, brain cells lose their normal functions and metabolism, and products of accumulated oxidative stress can accelerate and increase clinical complications in stroke patients [18].

In our study, protein oxidative damage has been related to an increased concentration of carbonyl groups in serum proteins in patients after ischemic and reperfusion injury. An increase in the concentration of this biomarker has been shown, both at the diagnosis of stroke < 4.5 h and on days 1 and 7 of thrombolytic therapy compared to the control group (Table 3). Protein damage was also found in the acute phase of stroke (<4.5 h) with the maximum concentration of carbonylated proteins on the first day after stroke and a slow decrease on the seventh day. Until now, the concentration of carbonyl proteins has not been determined during onset-to-needle, as other authors assessed the degree of protein damage only after 24 h [14,19,20,21]. Therefore, this is the first study of this type. Nevertheless, there have been reports of data where the marker of protein oxidative damage remained unchanged in all experimental groups describing the concentration of the biomarker after 24 h, 7 days and 3 months since the onset of stroke [12,22]. On the other hand, in the case of the concentration of melatonin’s metabolite in the urine samples of patients with ischemic stroke, the levels were lower compared to its levels in the urine of the control group (Table 3). This could suggest that patients treated with thrombolysis have impaired urinary excretion of the metabolite and low antioxidant protection. Similarly, a reduced level of serum melatonin and its metabolites in the urine of patients with stroke has been confirmed by other researchers [23,24,25,26,27]. In our study, an increase in the concentration of the melatonin metabolite was observed within the first 24 h compared to its level before <4.5 h in patients treated with thrombolytic therapy. The increased level of the metabolite was maintained over the following days. This may indicate the neuroprotective role of melatonin already present in the acute phase of stroke [15,24,25]. High levels of pineal-derived melatonin in human cerebrospinal fluid (CSF) protect against oxidative stress and act as a direct free radical scavenger with the ability to detoxify both reactive oxygen and reactive nitrogen species. The studies also revealed a positive statistically significant correlation between the concentration of melatonin measured at <4.5 h and 24 h and the concentration of carbonyl groups determined on the seventh day. On the other hand, a negative correlation was found between the concentration of carbonyl groups and the level of melatonin in <4.5 h, which may suggest gradual damage of proteins. The maximum production of reactive oxygen species can probably be reached only ≥24 h. Therefore, higher serum melatonin concentrations could be attributed to the compensation of oxidation products. The endogenous antioxidant system has been activated to maintain redox homeostasis [28], which may cause an increase in endogenous melatonin as a secondary factor of primary brain injury. In turn, in animal models of ischemic stroke, it was found that the supply of melatonin has a beneficial effect and reduces inflammation, apoptosis, mitochondrial dysfunction, cerebral oedema, cerebral infarction volume, and neurological dysfunction, contributing to increased survival [29,30,31]. It has also been observed that the level of 6-sulphatoxymelatonin in daily urine < 4.0 ng mL^−1^ can be considered a biological marker of cognitive impairment with memory loss [32], and the borderline value of serum melatonin < 51.5 pg mL^−1^ measured may be associated with the presence of ischemic stroke (OR = 3.12; *p* = 0.0463). On the other hand, a decrease in melatonin levels by 1.0 pg mL^−1^ may be associated with an increase in the probability of stroke [27].

In our study, it has been illustrated that the concentration of carbonyl groups before intravenous thrombolysis was lower in the subgroup of patients with a favourable functional result compared to the group of patients with a negative functional result (mRS: 3–6 points) in admission, discharge, after 3 months and after a year since the stroke. In addition, statistically significant differences were also observed in the concentration of the melatonin metabolite assessed on the 1st day of stroke for patients with mRS 0–2 points compared to the results of patients with mRS 3–6 points at the time of admission to the hospital (Table 2). Moreover, it was shown that a higher concentration of carbonyl groups in the acute phase of stroke was associated with a worse initial and late prognosis in patients treated with intravenous thrombolysis. There was a correlation between the levels of carbonyl groups measured at the time of onset by the NIHSS on admission and discharge and mRS also assessed on admission, and discharge, but also in the third month and after 1 year since the stroke (Table 4). A relationship was also found between 6-hydroxymelatonin sulphate assessed after 24 h and the scale of the severity of the neurological deficit on admission. Therefore, a relationship was observed between the levels of carbonyl groups and 6-hydroxymelatonin sulphate and the neurological disorders. It has been shown that the exanimated biomarkers can act as prognostic factors in the treatment of thrombolytic AIS. Carbonyl groups seem to be a particularly promising biomarker, as they can be considered a potential marker of the extent of hyperacute brain injury and its ultimate clinical consequences. Our findings show, for the first time, a significant correlation between the degree of oxidation of amino acid residues and 6-hydroxymelatonin sulphate in serum proteins, as well as in a modified Rankine scale assessing the functional status of patients after stroke. Other authors have so far managed to assess the relationship between the level of carbonyl groups and the geriatric depression scale [20]. Post-stroke depression is associated with increased disability and increased cognitive impairment, leading to a delay in the rehabilitation process. A positive correlation of carbonyl groups was also found with an increase in the volume of the lesion after DWI (diffusion-weighted imaging, R = 0.444, *p* = 0.05), described based on diffusion magnetic resonance imaging [11]. It has also been demonstrated that elevated levels of carbonyl groups and ox-LDL (oxidised low-density lipoprotein) groups increase thromboembolic oedema (TAO) in patients with Buerger’s disease [33]. Although there is information in the literature based on which it is believed that melatonin (*N*-acetyl-5-methoxytryptamine) and its metabolites may be involved in modulating oxidative stress in acute stroke [24], its levels in patients with ischemic stroke treated with thrombolysis have not been fully identified [23,27]. Therefore, there is no information about melatonin’s role in the long-term prognosis of hospitalised strokes. According to recent reports, the authors only managed to show that serum melatonin can be considered an independent predictive factor in the functional outcome after aneurysmal subarachnoid haemorrhage (aSAH) [34].

So far, no information on the content of these markers for individual stroke subtypes based on the TOAST (The Trial of ORG 10172 in Acute Stroke Treatment) classification has been provided. Moreover, the current reports focus mainly on a homogeneous group of respondents concerning patients with mainly atherosclerotic stroke [11]. In turn, our research was based mainly on lacunar stroke where we found lower values of carbonyl group concentration in proteins and an increase in the concentration of 6-hydroxymelatonin sulphate compared to other types of strokes. In contrast, during hospitalisation, critically ill patients show lower levels of melatonin compared to the group of patients with lacunar stroke. In extensive cortical lesions, there may be a delay in melatonin secretion during the first days after stroke [35]. Disturbed antioxidant status may lead to the progression of atherosclerotic lesions and an increased risk of cerebrovascular changes [27]. Also, the resulting neurological deficits could seriously impair endogenous melatonin secretion. Melatonin levels may also have been reduced due to its use as a scavenger for both oxygen and nitrogen reagents [36,37,38,39], which are elevated during ischemia or reperfusion. Also, the massive release of glutamate into the brain and its diffusion into the hypothalamus disrupts the clock neurons of the hypothalamus. The timing of glutamate release and its spread to the hypothalamus may vary, resulting in individual-specific disturbances in melatonin secretion [13]. It is also worth mentioning that abnormalities in the circadian rhythm of post-stroke patients may be quantitatively related to the distance of the stroke from the hypothalamus. However, direct damage to the suprachiasmatic nuclei (SCN) and SCN of the hypothalamus cannot be ruled out and neither can the destruction or blocking of the neuronal activity causing desynchronization of rhythmic processes and disturbance of melatonin secretion [13,35].

Interestingly, we have also identified a relationship between the values of carbonyl and 6-sulphatoxymelatonin groups and mortality in stroke patients. Lorente et al., studying patients with a stroke of the central cerebral artery found a correlation between melatonin-elevated blood serum level (>2.93 pg mL^−1^) and the mortality of patients [40]. Elevated melatonin levels could be the result of compensation for maintaining the balance between the oxidant and antioxidant states. Insufficient concentration of melatonin may lead to increased peroxidation and eventually the death of patients.

However, it should be remembered that this report has also some limitations. Most importantly, urine samples for testing melatonin levels were not collected from all patients at the same time of day, but were collected at the time of diagnosis. Moreover, it is single-centre research with a limited size of study population. The need to express informed consent for the collection of additional blood and urine samples for the determination of oxidative stress markers resulted in the exclusion of patients with aphasia or with impaired consciousness. Consequently, only patients with mild and moderate neurological deficits were enrolled in the study.

## 4. Materials and Methods

### 4.1. Study Group

The study included 123 patients aged 35 to 86 years with acute ischemic stroke who were admitted to the Department of Neurology of the Nicolaus Copernicus University Hospital in Bydgoszcz.

Patients undergoing combination therapy with diagnosed infections, neoplastic diseases, coronary episodes, kidney, and liver damage. Invasive procedures were excluded from the study. Patients with a history of cerebral infarction and inflammatory and autoimmune diseases were treated with steroids. The research was approved by the Bioethics Committee in Bydgoszcz, operating at the CM UMK in Toruń (No. KB 637/2016).

The Department of Neurology, University Hospital in Bydgoszcz, complies with Polish national standards. Patients were diagnosed with stroke following the criteria of ICD 10. Strokes were confirmed based on clinical evaluation and computed tomography (CT) performed at 22 to36 h after thrombolysis. Coexisting diseases were diagnosed according to the current guidelines of the European Society of Cardiology (ESC); The Diabetes Association; the Polish Section of Neurosonology, and renal dysfunction with estimated glomerular filtration rate < 60 mL/min/1.73 m^2^. Stroke classification was carried out under the commonly accepted classification of the aetiology of ischemic stroke TOAST.

The control group, which was considered healthy, consisted of 35 people aged 27–66 years. Their health status was assessed based on an interview and clinical evaluation performed by qualified medical personnel. In these people, no cardiovascular diseases, autoimmune diseases or cancer have been detected. The study did not include people with infections and other diseases or taking any medications, which could affect the final results. The patients’ consent to participate in the research was voluntary.

### 4.2. Biochemical Testing

The biological material for the study was collected by qualified medical personnel upon admission of patients to the Department of Neurology within <4.5 h and on days 1 and 7 after a stroke.

Blood samples were collected from the posterior vein in 5 mL tubes with a clot activator and a gel separator. Serum blood was centrifuged (3000× *g* for 15 min) and all samples were stored at −80 °C until protein carbonyl groups were determined U/mL. Additionally, urine samples were collected from each patient for the determination of 6-hydroxymelatonin sulphate. The content of 6-sulphahydroxymelatonin in human urine fluctuates in line with the 24 h concentration of melatonin in urine, plasma, and saliva. Melatonin shows a circadian rhythm of secretion, usually starting to rise between 6 p.m. and 8 p.m., peaking between 24 a.m. and 5 a.m., followed by a rapid decrease. All patients who urinated were kept in the same lighting conditions, the windows of the hospital neurology ward were darkened by the blinds.

Urine melatonin levels and serum protein carbonyl concentration were measured by ELISA using a commercial Immuno Biological Laboratories kit (IBL International GMBH, Hamburg, Germany) for melatonin (detection limit 0.1 pg/mL) and Immundiagnostik AG (Bensheim, Germany) for groups carbonyl (U/mL). Biochemical analyses were performed in accordance with the manufacturer’s recommendations.

### 4.3. Statistical Methods

A retrospective data analysis was performed in the study. The R software language (version 4.1.2) was used for descriptive statistical analysis.

The continuous variables were analysed for normal distribution and equality of variance.

The Shapiro-Wilk test was used to check the normality of the distribution of continuous variables. For one-dimensional analysis of continuous variables, Mann-Whitney U tests were used. The chi-square test was used to compare binary variables and the Spearman’s rank test was used to evaluate correlations. For statistically significant values, the *p*-value was <0.05. ROC, curves were also used for the statistical analysis, and cut-off points for the tested biomarkers were determined.

## 5. Conclusions

In summary, it should be emphasized that the obtained results may indicate increased oxidative stress, and the concentration of carbonyl groups may be a potential parameter of protein damage in thrombolytic patients with acute ischemic stroke. Something that should also be noticed is the important neuroprotective function of melatonin in the acute and subacute phases in patients treated with thrombolytic therapy. Melatonin supplementation can effectively prevent both behavioural and neurophysiological defects caused by cerebral hypoxia and ischemia, or spinal cord injury. Hormone supplementation will help us understand the interactions between exogenous melatonin and its endogenous pace and how these relationships can affect the outcomes of stroke patients. It is known that *N*-acetyl-5-methoxytryptamine has long been used in the alleviation of sleep disorders, shows no toxicity in humans, and its positive effects assessed in animal models have been noticeable [8,28].

Therefore, attention should be paid to designing and conducting clinical trials that will thoroughly investigate the safety and potential use of melatonin in stroke patients. Considering the above limitations, the research should be continued and, what is more important, it should be conducted in a larger population, including patients with severe symptoms of stroke treated endovascularly. An assumption can be made that melatonin may be additionally supplemented in thrombolytic therapy, but this requires confirmation in the future. In addition, the dependence on endogenous melatonin and night lighting in the intensive care unit (ICU) should also be considered.

## Data Availability

Data available from the authors.

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
