# Peer review of "Increased Oxidative Stress Markers in Acute Ischemic Stroke Patients Treated with Thrombolytics"

_ijms, 2022, doi:10.3390/ijms232415625_

Round 1

Reviewer 1 Report

The Manuscript presents research data supporting the idea that oxidative stress is one of the main factors that result in structural and functional disorders during brain stroke. The authors used the determination of the protein carbonyl groups in the blood samples as a sign of oxidative stress. The authors measured the content of 6-sulphahydroxymelatonin in human urine, a metabolite of melatonin known as an intrinsic antioxidant. The biochemical parameters were measured within 4.5 hours and on days 1 and 7 after a stroke. The authors concluded that the obtained results might indicate increased oxidative stress, and the concentration of carbonyl groups may be a potential parameter of protein damage in thrombolytic patients in acute ischemic stroke and suggested that melatonin functions as an essential neuroprotector of in the acute and subacute phases in patients treated with thrombolytic therapy.

The data supporting the above conclusions are presented as a statistical analysis of the two biochemical parameters. However, carbonylated proteins in the sera of patients and control individuals are highly unspecific and do not necessarily prove their origin from the brain. Determining F2-isoprostanes would be a much more specific indication of oxidative stress [1]. The content of melatonin metabolites in the urine depends on the individual circadian rhythm, and darkened blinds do not necessarily reproduce the circadian rhythm.

In addition, it is currently known that melatonin, a protein, and other antioxidants, do not protect mitochondria and cells from the protonated form of superoxide radical [2], perhydroxyl radical. Under conditions of stroke and hypoxia, there is local acidification of the affected tissue, which promotes the protonation of superoxide radical, which promotes isoprostane lipid peroxidation with the formation of F2-isoprostanes

Thus the presented work has a poor design and lacks a discussion of alternative pathophysiological mechanisms of stroke and clinical data.

Rferences

1.     Morrow JD, Roberts LJ (1996). "The isoprostanes. Current knowledge and directions for future research". Biochem. Pharmacol. 51 (1): 1–9. doi:10.1016/0006-2952(95)02072-1

2.     Barja  (2014) Chapter 1. The Mitochondrial Free Radical Theory of Aging. Prog Mol Biol Transl Sci. 2014; 127:1-27. doi: 10.1016/B978-0-12-394625-6.00001-5

Author Response

We would like to thank the Reviewer for the comments.

Responses to the review have been attached in the pdf file.

Reviewer 2 Report

The manuscript from Pawluk et al. shows an interesting research work on the use of carbonyl content and melatonin metabolite as biomarkers of AIS patient evolution. The authors conclude that patients after ischemic and reperfusion show a different evolution profile of protein oxidative damage based on the results obtained for those biomarkers from serum and urine samples. Carbonyl concentrations were higher in patients in the acute phase (first 24h) in comparison with those one found in the incoming days. Regarding the 6-hydroxymelatonin sulphate, concentrations were lower in AIS patients than those in the control.

In summary, the results shown in this work could shed light on the development of new therapies for AIS treatment.

This manuscript can be accepted for publishing in IJMS.

Author Response

We would like to thank the Reviewer for the valuable remarks and comments, which significantly helped us to improve our manuscript. Below we present point-by-point responses to the comments.

The manuscript from Pawluk et al. shows an interesting research work on the use of carbonyl content and melatonin metabolite as biomarkers of AIS patient evolution. The authors conclude that patients after ischemic and reperfusion show a different evolution profile of protein oxidative damage based on the results obtained for those biomarkers from serum and urine samples. Carbonyl concentrations were higher in patients in the acute phase (first 24h) in comparison with those one found in the incoming days. Regarding the 6-hydroxymelatonin sulphate, concentrations were lower in AIS patients than those in the control.

In summary, the results shown in this work could shed light on the development of new therapies for AIS treatment.

This manuscript can be accepted for publishing in IJMS.

Thank you very much for your positive opinion on the submitted manuscript.

Minor linguistic corrections have been made.

                                                                                   with kindest regards

                                                                                          Hanna Pawluk              

Reviewer 3 Report

This manuscript entitled “The contribution of oxidative stress in patients with acute ischemic stroke treated with thrombolytics” is a clinical study. The authors aim to evaluate the progressive profile of the melatonin metabolite, 6-hy-droxymelatonin sulphate (6-SM), in the urine as well as to assess carbonyl groups in the blood of patients with acute ischemic stroke treated with intravenous thrombolysis. The  research also focused on the comparison of the obtained results with the functional status of patients assessed according to the neurological scales and with a control group of healthy people. The study is interesting and well-written. There are some aspects need to be addressed before publication.

1.     The contribution of oxidative stress to ischemia is widely recognized. The title of the study should be revised and more specific.

2.      It seems there are only two groups in the study. The authors may need to considering adding a group including the patient who do not receive the thrombolytics.

3.   The authors should divide the results into several segments using subheadings. In the current version, it is difficult to read. Also, for the discussion, there are too many separate paragraphs, the authors should consider to consolidate them into several sections.

Author Response

We would like to thank the Reviewer for the valuable remarks and comments, which significantly helped us to improve our manuscript. Below we present point-by-point responses to the comments.

This manuscript entitled “The contribution of oxidative stress in patients with acute ischemic stroke treated with thrombolytics” is a clinical study. The authors aim to evaluate the progressive profile of the melatonin metabolite, 6-hydroxymelatonin sulphate (6-SM), in the urine as well as to assess carbonyl groups in the blood of patients with acute ischemic stroke treated with intravenous thrombolysis. The  research also focused on the comparison of the obtained results with the functional status of patients assessed according to the neurological scales and with a control group of healthy people. The study is interesting and well-written. There are some aspects need to be addressed before publication.

The contribution of oxidative stress to ischemia is widely recognized. The title of the study should be revised and more specific.

The title of the manuscript has been changed to:

“Effect of selected parameters of oxidative stress on the functional outcome of patients with ischemic stroke”.

It seems there are only two groups in the study. The authors may need to considering adding a group including the patient who do not receive the thrombolytics.

Thank you very much for pointing out the suggestion to conduct research on another research group including patients not treated with thrombolysis. We are currently conducting research on such a group of patients and in the near future we will try to conduct a comparative analysis of the results obtained and publish them.

The authors should divide the results into several segments using subheadings. In the current version, it is difficult to read. Also, for the discussion, there are too many separate paragraphs, the authors should consider to consolidate them into several sections.

Thank you for your suggestion. Corrections have been included in the manuscript.

                                                                                   with kindest regards

                                                                                         Hanna Pawluk                                                                    

Round 2

Reviewer 1 Report

In the future, I would recommend to respected authors to consider not one rather indirect marker of oxidative stress, but at least two or three other vore direct parameters reflecting oxidative stress. Currently, there are many, and the selection depends, of course, on the availability and quality of equipment. I would also recommend updating your knowledge on oxidative stress. Many  of the previous paradigms became outdated and even obsolete

Author Response

Comments to the Reviewer

Thank you very much to the Reviewer for the review of our article. We are grateful for pointing out valuable comments and for sending important information on oxidative stress.

We will try to further deepen our knowledge based on the latest scientific sources.

We will also pay attention to the importance of the biochemical parameters indicated by the Reviewer, along with their implementation in future research. Undoubtedly, the indicated parameters will better reflect the ongoing process of oxidative stress, which accompanies the etiology of many free radical diseases, including ischemic stroke.